# ARE LLMs BETTER THAN REPORTED? DETECTING LABEL ERRORS AND MITIGATING THEIR EFFECT ON MODEL PERFORMANCE

## ABSTRACT

NLP benchmarks rely on standardized datasets for training and evaluating models and are crucial for advancing the field. Traditionally, expert annotations ensure high-quality labels; however, the cost of expert annotation does not scale well with the growing demand for larger datasets required by modern models. While crowd-sourcing provides a more scalable solution, it often comes at the expense of annotation precision and consistency. Recent advancements in large language models (LLMs) offer new opportunities to enhance the annotation process, particularly for detecting label errors in existing datasets. In this work, we consider the recent approach of LLM-as-a-judge, leveraging an ensemble of LLMs to flag potentially mislabeled examples. Through a case study of four datasets from the TRUE benchmark, covering different tasks and domains, we empirically analyze the labeling quality of existing datasets, and compare expert, crowd-sourced, and our LLM-based annotations in terms of agreement, label quality, and efficiency, demonstrating the strengths and limitations of each annotation method. Our findings reveal a substantial number of label errors, which, when corrected, induce a significant upward shift in reported model performance. This suggests that many of the LLMs so-called mistakes are due to label errors rather than genuine model failures. Additionally, we discuss the implications of mislabeled data and propose methods to mitigate them in training to improve model performance.

## 1 INTRODUCTION

Natural Language Processing (NLP) benchmarks have long served as a cornerstone for advancing the field, providing standardized datasets for training and evaluating methods and models (Wang et al., 2019; Hendrycks et al., 2021; Srivastava et al., 2023). These datasets have been developed over the years for various tasks and scales, annotated using different schemes. Initially, human domain expert annotation was preferred, as these experts possess the skills necessary to determine correct labels accurately. However, as models have increased in size (Devlin et al., 2019; Raffel et al., 2020; Brown et al., 2020), the demand for larger datasets has also grown (Kaplan et al., 2020). Since expert annotation is cost-prohibitive, it does not scale well to meet these new demands. The demand for large quantities of annotated data quickly and cost-effectively has led researchers to adopt crowd-sourcing, often sacrificing expertise for scale.

That way or another, constructing datasets heavily involves making compromises in annotation, trading off between scale, efficiency and expertise. Even when annotated by experts, datasets can naturally contain labeling errors, arising from factors such as task subjectivity, annotator fatigue, inattention, insufficient guidelines, and more (Rogers et al., 2013; Reiss et al., 2020; Sylolypavan et al., 2023). Mislabeled data is even more pronounced when non-expert annotators are involved (Kennedy et al., 2020; Chong et al., 2022b). Widespread mislabeled data is particularly concerning because both the research community and the industry rely heavily on benchmarks. In training data, label errors harm model quality and hinder generalization, while in test sets, they lead to flawed comparisons, false conclusions, and prevent progress.

Recent advancements in large language models (LLMs) (Ouyang et al., 2022; Chiang & Lee, 2023; Li et al., 2023; Gat et al., 2024) present new opportunities to improve the annotation process, specifi-

cally in detecting label errors within existing datasets. Rather than re-annotating entire datasets (e.g., through experts or crowd-workers), we consider the recent approach of LLM-as-a-judge (Zheng et al., 2023), and propose a simple yet effective method by leveraging an ensemble of LLMs to flag a set of potentially mislabeled examples. These can then be sent to experts for re-annotation and correction, or even get filtered during training.

Specifically, we construct an ensemble model using multiple LLMs with diverse prompts, gathering both their predicted labels and corresponding confidence scores. These predictions are contrasted with the original labels, and instances where the LLMs *strongly disagree* with the original label (i.e., show high confidence in a different label) are flagged as potential mislabeling cases. Additionally, we not only explore the role of LLMs in detecting errors but also evaluate their performance as annotators, comparing them with expert and crowd-sourced annotations. We assess these approaches in terms of agreement, label quality, and efficiency, highlighting their strengths and limitations.

We aim to answer the following questions through a comprehensive end-to-end study: (1) To which extent current benchmarks include mislabeled data? (2) Can LLMs detect label errors? (3) How do expert, crowd-sourced, and LLM-based annotations compare in quality and efficiency? and (4) What are the implications of mislabeled data on model performance and can we mitigate their impact?

To this end, we choose the TRUE benchmark (Honovich et al., 2022) – A collection consolidating 11 existing datasets annotated for factual consistency in a unified format – as a case-study and empirically investigate its labeling quality. Specifically, we analyze four datasets from TRUE with binary factual consistency annotation originating from different tasks. This enables us to explore multiple tasks and domains while benefiting from a uniform labeling schema.

This paper presents both methodological and empirical contributions. We propose a straightforward approach for detecting potential mislabeled examples, revealing a substantial number of label errors in existing datasets, ranging from 6% to 21%. Additionally, we demonstrate that the precision of LLMs in identifying errors improves with their confidence in an incorrect label; when their confidence exceeds 95%, over two-thirds of those labels are, in fact, errors. Moreover, we show that LLM-based annotations not only excel in error detection but also perform similarly to, or better than, traditional annotation methods, offering better trade-offs between quality, scale, and efficiency. Finally, we empirically illustrate the negative impact of mislabeled data on model training and evaluation. We propose a simple, fully automated method for addressing label errors, improving the performance of fine-tuned models by up to 4%. In evaluation, we found that mislabeled data can significantly distort reported performance; LLMs may perform up to 15% better. This indicates that many so-called prediction errors are not genuine errors but are instead human annotation mistakes.

## 2 DATA ANNOTATION APPROACHES

### 2.1 TRADITIONAL ANNOTATION APPROACHES

**Crowd-Sourcing.** Crowd-sourcing has been widely used to annotate large-scale NLP datasets (Rajpurkar et al., 2016; Williams et al., 2018; Wang et al., 2022) because it enables the rapid collection of labeled data at scale. However, the reliability of crowd-sourced annotations has been questioned, as quality control remains a challenge, with labeling inconsistencies becoming more frequent as dataset complexity increases (Lu et al., 2020; Allahbakhsh et al., 2013). One of the key advantages of crowd-sourcing has traditionally been its ability to handle tasks requiring human creativity or subjective judgment — areas where models have historically struggled. However, this advantage is fading, as models now approach near-human performance on such tasks (Chiang & Lee, 2023; Chen & Ding, 2023), and crowd workers increasingly rely on models for assistance, diminishing the human element in the process (Veselovsky et al., 2023b;a).

**Human Experts.** Expert annotation is a reliable approach for NLP tasks that require domain-specific expertise (e.g., medical or legal domains) and for tasks that demand deep cognitive engagement, such as those requiring training to understand complex guidelines or intelligent, attentive annotators. However, this approach is slow and expensive compared to crowd-sourcing (Snow et al., 2008; Chau et al., 2020), limiting its scalability for the large datasets needed to train modern LLMs. Managing the trade-off between cost-effectiveness and annotation quality is critical, especially in tasks that require domain-specific accuracy (Chau et al., 2020). Maintaining inter-annotator agree-

ment among experts presents an additional challenge, further driving up costs (Baledent et al., 2022). Hybrid approaches, combining expert and crowd-sourced annotations, can mitigate this trade-off, though expert involvement remains essential for high-quality labels (Nguyen et al., 2015).

## 2.2 LLM AS AN ANNOTATOR

As shown in recent studies (Gilardi et al., 2023; Kholodna et al., 2024; Li et al., 2023), LLMs can be integrated into the annotation process, as they are fast, relatively cheap, and obtain decent performance. Despite the promise of LLMs acting as annotators, LLMs make mistakes, and therefore their annotations can not be considered as Gold labels (Chen et al., 2024; Bhat & Varma, 2023). Still, incorporating LLMs in the labeling process offers cost and scalability benefits.

In this study we propose to utilize LLM annotations, alongside with their confidence, in order to automatically detect errors in existing labeled datasets. Specifically, we propose a general schema for re-classification via LLMs, described as follows. We re-label the dataset via LLM, and obtain a predicted probability for each class. When using a model with access to the parameters, these probabilities can be extracted directly at inference time. When using models available via public APIs, probabilities may be given as token log-probabilities (if available through the API) or approximated via sampling. After annotating via LLMs, examples for which there is a *strong disagreement* between the LLM annotation and the original label (i.e., high LLM probability for another label), are flagged as potentially mislabeled. If in a test set, the flagged examples could then be re-examined by human experts to determine their true label. If in a training set, the same solution may apply, but such examples could also be filtered or have their label changed according to the LLMs prediction.

To overcome the variance in LLM-generated answers, we suggest averaging over various prompts, and over different models, obtaining one strong LLM ensemble. As shown in Appendix A, this approach not only reduces variance but also enhances model performance and detection capabilities.

## 3 EXPERIMENTAL SETUP

### 3.1 DATA

As a case-study, we choose to explore the extensive and widely used TRUE benchmark (Honovich et al., 2022), which is typically used as an evaluation set (Steen et al., 2023; Gekhman et al., 2023; Wang et al., 2024; Zha et al., 2023). It consists of 11 datasets from various NLP tasks such as summarization and knowledge-grounded dialogue. This benchmark is unique in its approach of bringing multiple datasets and tasks into a unified schema of binary factual consistency labels. Each dataset is transformed from its original structure (e.g., a source document and a summary) into two input texts, *Grounding* and *Generated Text*, and a binary label indicating whether the generated text is factually consistent w.r.t the grounding. This enables us to examine multiple tasks and domain under the same umbrella at once, while maintaining a unified binary-label schema.

Specifically, we focus on four TRUE datasets, one from each task (summarization, dialogue, fact verification, paraphrasing). Each of these datasets have been annotated with different guidelines, for a different purpose, and with a slightly different annotation procedure:

**MNBM (Maynez et al., 2020): Summarization.** This dataset provides annotations for hallucinations in generated summaries from the XSum dataset (Narayan et al., 2018). *Grounding* refers to the source document that the summary is based on, while *Generated Text* consists of model-generated summaries, which may include hallucinated information not present in the source. Three human annotators, trained for the task through two pilot studies, annotated the dataset for the existence of hallucinations. In TRUE, the binary annotations were determined by majority vote.

**BEGIN (Dziri et al., 2022): Dialogue.** This dataset evaluates groundedness in knowledge-grounded dialogue systems, where responses are expected to align with an external *Grounding* source, typically a span from Wikipedia. *Generated Text* refers to model-generated dialogue responses that were fine-tuned on datasets like Wizard of Wikipedia (Dinan et al., 2019). Data was annotated into entailment/neutral/contradiction labels, by three human annotators, trained for the task through two pilot studies, aggregated by majority vote. In TRUE, binary annotations were then determined by the entailment/not-entailment partition.

**VitaminC (Schuster et al., 2021): Fact Verification.** This dataset is based on factual revisions of Wikipedia. The evidence, or *Grounding*, consists of Wikipedia sentences, either before or after these revisions. Most human involvement came from creating *Generated Text* rather than the annotation process, with annotators writing claim/evidence pairs derived from Wikipedia revisions, inherently generating labeled data for fact verification. Synthetic examples from the FEVER dataset (Thorne et al., 2018) were also included. Additionally, three annotators reviewed 2,000 examples, presumably to ensure data quality.

**PAWS (Zhang et al., 2019): Paraphrasing.** This dataset consists of paraphrase and non-paraphrase pairs. *Grounding* refers to source sentences drawn from Quora and Wikipedia, while *Generated Text* was automatically generated through controlled word swapping and back-translation. Five human annotators annotated the dataset with binary labels w.r.t paraphrasing correctness. The dataset includes both high- and low-agreement annotations.

For each of the four datasets, we randomly sampled 1000 examples (or the whole dataset if the number of examples is smaller than 1000). These examples are annotated via LLMs as described in subsection 2.2. We set an evaluation (i.e., test set) based on 160 randomly sampled examples from each dataset (a total of 640), while the rest remain for training and validation (they will be relevant for subsection 6.1). In addition to the LLMs annotation, the evaluation set is also re-annotated by human experts, which are two of this paper's authors, fully familiar with the task, and by three human annotators per example via crowd-sourcing.

### 3.2 ANNOTATION PROCEDURE

This subsection outlines the annotation procedures for the various approaches. For additional implementation and technical details not covered here, please refer to the Appendix B.

**LLMs.** We follow the general schema described in subsection 2.2 by utilizing LLM labels with their confidence for each class to detect mislabeled data. To this end, we annotate the data with four different models: GPT-4 (OpenAI, 2023), PaLM2 (`text-bison@002`) (Anil et al., 2023), Mistral (7B)[1] (Jiang et al., 2023) and Llama 3 (8B)[2] (Dubey et al., 2024). We designed four different prompts, to control the variance caused by task description, and report more stable results. The prompts are designed as a zero-shot classification task: the requested output is a single token, either `'0'` for factual inconsistency or `'1'` for factual consistency. Instead of taking the binary output, we extract the corresponding probability from logits or log-probabilities, as estimation of the model's confidence for the predicted class. Overall, for each example we have $4 \times 4$ probabilities for binary labels $P_{model}^{prompt}(y = 1|x)$. As GPT-4 and PaLM2 showed better performance (i.e., higher ROC AUC) and consistency (i.e., higher IAA), in the following sections we denote their average probabilities as the single *LLM* $p$ where $p = P(y = 1|x)$ or *LLM annotation* in the binary case $\mathbb{I}\{p > 0.5\}$.

**Crowd-sourcing.** We utilize the platform of Amazon Mechanical Turk (MTurk) to recruit crowd-workers for annotating 100 examples from each dataset (a total of 400), and to design the interface layout. Examples were randomly assigned to annotators. Each annotated example was manually reviewed. Rejected examples were returned to the pool and re-annotated, until each example had been annotated by three different annotators. To prevent (as much as possible) LLM use in the crowd-sourcing annotation, we disabled the possibility of right-click and `Ctrl+c` in the platform (as suggested by Veselovsky et al., 2023a). To obtain a single label per example, we consider two different aggregations: (1) *Majority* - by majority vote, and (2) *Strict* - if any annotator marks it *inconsistent*, that becomes the label.

**Experts.** All examples where the LLMs' annotations differed from the original label, regardless of the LLMs' confidence, were annotated by human experts. These experts were two of this paper's authors, who are fully familiar with the guidelines and task characteristics. Each example was annotated by both annotators independently on a scale of 0 (*inconsistent*) to 1 (*consistent*). Examples were shuffled and did not appear in any specific order, and neither the original nor LLM labels were presented, just the plain texts. Subsequently, on the examples where the annotators did not agree with each other, a reconciliation phase took place, where both annotators discussed, attempting to resolve the disagreement. After re-annotating all the conflicted examples, we consider the *Gold*

---

[1] `https://huggingface.co/mistralai/Mistral-7B-Instruct-v0.2`
[2] `https://huggingface.co/meta-llama/Meta-Llama-3-8B-Instruct`

Table 1: Example of an annotation error in the original datasets, discovered by LLMs and corrected by experts. In Appendix Table 8 we provide additional examples.

---

**Dataset:** BEGIN

**Grounding:** Hillary Clinton, the nominee of the Democratic Party for president of the United States in 2016, has taken positions on political issues while serving as First Lady of Arkansas (1979–81; 1983–92), First Lady of the United States (1993–2001); **Generated Text:** She is the nominee in 2016.

**Original Label:** 0    **LLM** $p$: 0.98    **Gold Label:** 1

**Explanation**: She (Hillary Clinton) is indeed the nominee in 2016 as specifically stated in the grounding.

---

Table 2: Summary of LLM disagreement and label error rates across different datasets. %pos is the percentage of positive (i.e., the *consistent* class) examples in the data. % LLM disagree refers to the percentage of examples where the LLM label differs from the original one. % error indicates the error rate in the sampled test set, while the number in parentheses denoting the estimated lower bound of the error rate for the entire dataset.

| Dataset | Task | % pos | % LLM disagree | % error |
|---------|------|-------|----------------|---------|
| MNBM | Summarization | 10.6 | 39.4 | 16.9 (11.6) |
| BEGIN | Dialogue | 38.7 | 34.4 | 21.2 (15.8) |
| VitaminC | Fact Verification | 52.5 | 17.5 | 8.1 (4.4) |
| PAWS | Paraphrasing | 44.3 | 22.5 | 6.2 (3.0) |

*label* to be either the original label, in case the LLM label agrees with it, or the experts resolution, in case of there was a disagreement between them.

## 4   LABEL ERROR ANALYSIS AND THE ROLE OF LLMs IN ERROR DETECTION

### 4.1   DO CURRENT BENCHMARKS INCLUDE MISLABELED DATA?

To address the first research question, we conducted a detailed analysis of current benchmarks, identifying the extent to which mislabeled data exists across various datasets. Following the procedure described in subsection 3.2, we annotate the test-set using LLMs. We then contrast these annotations with the original labels, to find disagreements. As shown in Table 2, the disagreement rate is significant and can be up to $\sim 40\%$ of the examples. For instance, for 34.4% of the examples in the BEGIN dataset, the LLM ensemble label differ from the original label. Example for such label error detection is presented in Table 1. While usually we would say that this means that the LLMs performed poorly, we choose to further investigate these examples and settle the disagreement. To this end, we asked human experts to re-annotate these examples (as described in subsection 3.2), without any knowledge on LLM's $p$ or the original label. After experts re-annotation, we can conclude which is correct: the original label, or the LLMs?

Our findings show a considerable number of label errors for all examined datasets (see the %error column in Table 2). Based on the experts *Gold* label and the sample sizes, we also estimate a lower bound for the total percentage of label errors in the full datasets. For this calculation, we employed the Clopper-Pearson exact method (Clopper & Pearson, 1934) to construct a 95% confidence interval for the binomial proportion, adjusted by a finite population correction (FPC) (see more details in Appendix, E.1). We provide the lower bound of these confidence intervals in parenthesis Table 2, in the %error column. For instance, we bounded the %error in MNBM to be at least 11.6%.

### 4.2   CAN LLMs AUTOMATICALLY DETECT LABEL ERRORS?

As described in subsection 4.1, we utilize LLMs to flag candidates for mislabeling, and indeed find label errors. In this subsection, we focus on the LLM viewpoint, including the effect of LLM confidence, and the power of ensemble (i.e., aggregating multiple LLMs). LLM annotations are valuable for flagging mislabeled data, offering more than just binary labels. By considering LLM confidence scores alongside their predictions, we can improve the precision of automatic error detection.

To better assess the utility of LLMs in detecting label errors, we break down the flagged examples into confidence-based bins. The rationale behind this is that not all flagged examples should be treated equally. Some instances are flagged with lower confidence, indicating that the LLM recognizes a potential issue but is uncertain about the label it suggests. On the other hand, when the LLM is highly confident in assigning a label that opposes the original one, it serves as a stronger signal of a possible label error.

Figure 1 shows the rate of the experts agreement with the LLMs compared to the agreement with original labels, divided into confidence-based bins. Each bin includes at least 35 examples, and is defined by a confidence interval of 95% based on bootstrap sampling (see Appendix E.2 for further

details). The bins reflect increasing levels of LLM confidence in its predicted label (i.e., a stronger disagreement between LLMs and the original labels).

From the figure, we observe a clear trend: as LLM confidence increases, so does its precision in detecting label errors in the original dataset. In the highest confidence bin, LLM annotations surpass the original labels in agreement with expert re-labeling, and this difference is statistically significant. This indicates that when the LLM is highly confident in its disagreement with the original label, the labeled example serves as a strong candidate for a labeling error. Note that even in cases where the expert agreement with LLMs were below 50%, mislabeled data was still discovered. Finally, we found that using multiple LLMs in an ensemble is important for detecting label errors. As the number of models increases, we achieve not only higher-quality labels but also improved error detection capabilities (see Appendix A for the relevant experiments).

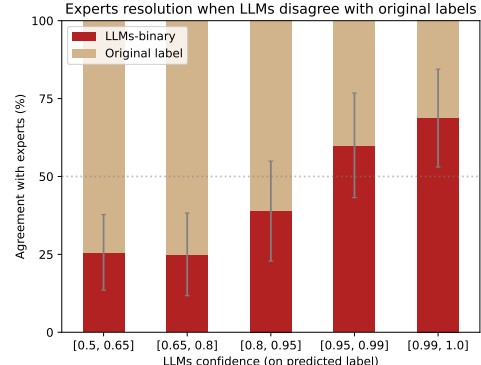

Figure 1: When LLMs disagrees with original labels - who is correct?. As the LLM's confidence grows, so does the precision of identifying an error in the original labels.

## 5 COMPARING ANNOTATION APPROACHES

As discussed in section 2, we focus on three main annotation approaches, each with its own benefits and drawbacks. These approaches differ in how they manage the trade-offs between label quality, scalability, and cost. In the following section, we discuss and compare their characteristics.

### 5.1 ANNOTATION QUALITY

When annotating or validating a dataset, one of our main concerns is the quality of the labels, or in other words, establishing a reliable gold standard. However, each annotation approach produces different labels. To estimate the quality of these approaches, we measure the agreement between different annotations using the weighted F1-score (which accounts for both classes). Note that this metric is not symmetric, meaning that treating one annotation as the *true* label and the other as the *prediction*, or vice versa, can result in different scores.

Figure 2 presents the F1-score between each pair of annotation approaches. As the figure shows, LLMs have disagreements with the *original* labels (0.72). Yet, as discussed in subsection 4.1, the original labels themselves contain mistakes, so this disagreement does not necessarily indicate poor performance of the LLMs. When considering the *Gold* as the true label, LLM performance increases to 0.83. This suggests that LLMs, despite their discrepancies with the original labels, perform closer to the truth than initially reported. The *Gold* label, obtained by experts, has high agreement with both the *Original* and *LLM* labels. On the other hand, the *MTurk-Majority* approach performs poorly, with near-random F1-scores compared to both the original and gold labels, and even when compared to its stricter variant, *MTurk-Strict*. The results indicate that basic crowd-sourcing, without additional training to enhance crowd-workers into specialized sub-experts, performs significantly worse compared to other approaches, including LLM-based methods.

**Crowd-sourcing.** For crowd-sourcing, the reported F1-score does not provide the complete picture. When we focus on individual annotators, we see that those who annotate more examples generally deliver higher-quality annotations, achieving greater accuracy when compared to both the original and gold labels (see Figure 3). This phenomenon can be explained by two hypotheses: (1) a learning process— as the annotators see more examples, they improve at the task, or (2) users who dedicate time to annotating multiple examples are likely those who either read the guidelines carefully and strive to perform the task to the best of their ability, or are naturally proficient at the task and therefore continue annotating. Even though annotators who label more instances tend to provide higher-quality annotations, they are less common—most annotators tend to stop after only a few examples. This distribution of annotators results in overall insufficient annotation quality. Pre-

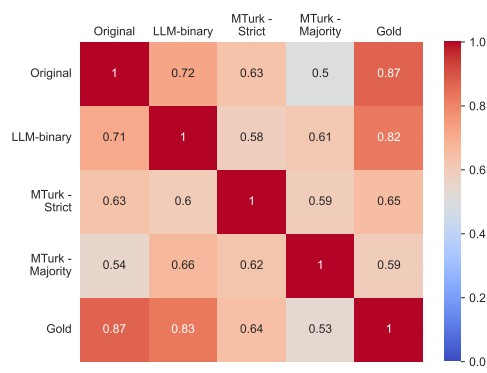 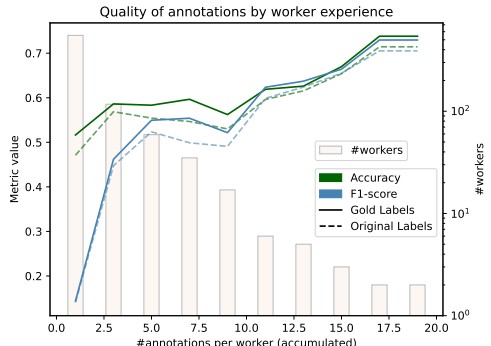

Figure 2: Comparison between all annotation methods, measured by the weighted-F1-score. Rows represent the *"true"* label and columns represent the *"prediction"*. For instance, the score of *LLMs* compared to the *Original* label is 0.72.

Figure 3: **(x-axis)** at list $x$ annotations per annotator. **(Right y-axis)** The number of annotators with at least $x$ annotations (bins). **(Left y-axis)** the average F1-score or accuracy for all user annotations with at least $x$ annotations.

qualification tests are often used to shift this distribution from the "average worker" towards more experienced or dedicated annotators; however, this requires a significantly larger budget and greater micro-management involvement from the researcher.

## 5.2 CONSISTENCY

Usually, when annotating a dataset, more than one annotator is involved. This applies to crowd-workers, experts, and even LLMs— in this study, we use an ensemble of different LLMs and prompts. The use of multiple annotators, similar to an ensemble, is meant to overcome the variance between individuals, which can arise from the subjective nature of NLP tasks, different interpretations of instructions, lack of experience, task difficulty, and cognitive bias (Uma et al., 2021).

As such, a common practice in the NLP community is to report Inter Annotator Agreement (IAA)— a set of statistical measures used to evaluate the agreement between individuals. Typically, IAA can be viewed as an adjustment of the proportion of pairwise agreements, where 0.0 indicates random agreement. We focus on Fleiss's $\kappa$ (Fleiss, 1971), as it accounts for label imbalance and multiple ($>$ 2) annotators. High IAA, or low variance between independent annotators, is considered an indicator of high-quality annotation. In Table Table 3, we report the agreement between annotators across different approaches. For LLMs, we report two variants: (1) same model, different prompts; and (2) different models, where each model's result is the aggregation across prompts. For reference, we also include the IAA from the original annotations, as reported in the original papers: *MNBM* reported an average Fleiss's $\kappa$ of 0.696 for the hallucination annotation task; *BEGIN* reported Krippendorff's $\alpha$ (a generalization of Fleiss's $\kappa$) of 0.7; *VitaminC* reported Fleiss's $\kappa$ of 0.7065 on a sample of 2,000 examples; and *PAWS* reported a 94.7% agreement between a single annotator's label and the majority vote on the Wikipedia subset used in TRUE.

Table 3: Inter-Annotator Agreement in different Annotator groups. %agreement is the proportion of pairwise annotator comparison. Fleiss's $\kappa$ (disagree. subset) refers to the $\kappa$ over the subset of disagreement between LLM and the original label.

| Annotator group | Fleiss's $\kappa$ | %agreement | #examples | Fleiss's $\kappa$ (disagree. subset) | #annotators |
|---|---|---|---|---|---|
| **Experts** | | | 222 | | 2 |
| Before reconciliation | 0.486 | 75.7 | | 0.486 | |
| After reconciliation | 0.851 | 93.2 | | 0.851 | |
| **MTurk** | 0.074 | 60.5 | 400 | -0.004 | 3* |
| **LLM (different prompts)** | | | 640 | | 4 |
| GPT-4 | 0.706 | 85.3 | | 0.571 | |
| PaLM2 | 0.750 | 87.7 | | 0.696 | |
| LLaMA3 | 0.219 | 71.7 | | 0.078 | |
| Mistral | 0.459 | 73.2 | | 0.314 | |
| **LLMs (different models)** | 0.521 | 77.5 | 640 | 0.389 | 4 |

**Experts.** While it's true that reconciliation naturally leads to increased agreement, the significant improvement in IAA we observed highlights its importance. Though this phase is less common in practice, it is crucial not only for increasing agreement but also for improving the overall quality of annotations and ensuring more reliable outcomes. Interestingly, label changes in this phase were not symmetric, as most changes (69.3%) were in the direction of *consistent* $\rightarrow$ *inconsistent*, where one annotator found an inconsistency that the other did not (see all change details in Appendix 6). It is important to note that the $\kappa$ obtained by the experts (both before and after reconciliation) was calculated on a more challenging subset, where the original label differed from the LLM prediction, and should be interpreted with this context in mind. This is reflected in the decrease in $\kappa$ observed for all other annotator groups on this subset.

**LLMs.** GPT-4 and PaLM2, the better-performing LLMs on this task, show high IAA, with $\kappa = 0.706$ and $\kappa = 0.75$, respectively, which is similar to the experts' reported $\kappa$. This suggests a comparable level of variance and quality in annotation, providing further empirical evidence for considering LLMs as annotators. This property adds to previous studies showing LLMs' quality as surrogates for human preferences (Zheng et al., 2023) or evaluations (Chiang & Lee, 2023).

**Crowd-Sourcing.** Crowd workers showed near-random agreement ($\kappa = 0.074$), indicating poor-quality annotations. Only 40.8% of the examples were labeled unanimously, while the rest included annotations from both classes. Even among the subset of examples unanimously labeled as *consistent*, 37.9% are labeled as *inconsistent* in both original and gold labels, pointing to a lack of attention and thoroughness. See more details on the crowd-sourced annotation distribution in Appendix B.1.2.

### 5.3 Cost and Scalability

LLM-based annotation is significantly cheaper and faster than crowd-sourcing platforms like MTurk, especially when considering the additional time required for human review cycles. It is estimated to be 100 to 1,000 times more cost-effective than using human annotators, including experts. This scalability and speed make LLMs a highly efficient alternative for large-scale annotation tasks.

## 6 Implications of Mislabeled Data

### 6.1 Training on Mislabeled Data

Training on mislabeled data can harm model performance and stability, as learning from errors makes it harder to identify consistent patterns. The impact depends on various factors , such as the fraction of mislabeled data and the training procedure. In this subsection, we show that addressing this issue, even heuristically, significantly improves model's performance on a test set.

**Handling Label Errors.** In order to handle label errors in the training set, and reduce its effect on model performance, we propose two manipulations. For both manipulations, we use similar methodology as discussed in this paper, and flag examples where the LLMs $p$ strongly disagree (i.e., above a certain confidence threshold) with the original label. The first manipulation is *filtering* flagged examples out, which maintains a "cleaner" yet smaller training set. The second manipulation is label *flipping* for flagged examples, which maintains the same amount of data, but may also cause harm if flipping too many correct labels.

**Experimental Setup.** We set the training set to be the additional data examples from the datasets (i.e., MNBM, BEGIN, VitaminC, PAWS), which are disjoint from the test set. Note that we posses gold labels for the test set alnoe, while for the training set we only extract $p$ via GPT4 and PaLM2. The finetuning procedure includes splitting the training set into train and validation sets, and fine-tuning on the train set. We report results averaged over five seeds.

As an ablation study, we also apply these manipulations on a random subset of examples rather than the flagged examples. TThe ablation study aims to maintain a consistent number of training examples, while the ablation for flipping aims to address the claim that in some cases, a relatively small fraction of label errors may be even considered as a noise that improves model robustness (e.g., as in label perturbation (Zhang et al., 2018) or label smoothing (Szegedy et al., 2016)).

---

*Multiple MTurk workers have participated in annotation, yet exactly 3 annotations per example were obtained. Annotators independence assumption was made to calculate Fleiss's $\kappa$ as with 3 annotators.

We conducted this experiment starting from two base models: `DeBERTa-base-v3`,[3] and a fine-tuned version of it on classic NLI datasets, which we will refer to as the NLI-base model[4]. We chose the NLI-base model as NLI tasks closely resemble factual consistency evaluation (FCE), making it well-suited for this experiment. Given the similar trends, we present the results for the NLI model here. Additional experiments and implementation details can be found in Appendix D.1.

**Results.** Figure 4 shows the results of our experiments. In our confidence-based approaches, we clearly see the trend that as the confidence threshold—according to which our manipulations are applied—grows, our manipulation results in improved ROC AUC for both models. This trend eventually (i.e., for high enough LLM confidence) brings these approaches to significantly outperform the baseline. In contrast, when we applied our manipulations on random subsets, we generally see a diminishing effect of manipulation, converging to the no-manipulation baseline.

Comparing between the handling approaches, it appears that flipping is better than filtering for high confidence. We hypothesize that this stems from the amount of data that remains after flipping (i.e., the same amount as before the flipping) compared to the filtering approach, combined with the high error rate in these datasets. Note that this is contrary to the random case where filtering is better than flipping, as flipping a subset with low error-rate brings more damage than value.

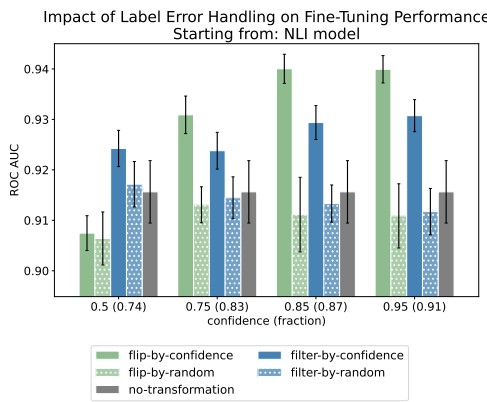

Figure 4: Fine-tuning a model on a transformed dataset. The gray bar is the original dataset - without any changes. The green bars present results for label flipping for a subset of examples, determined by LLMs-confidence (plain), or at random (dotted). The blue bars represent filtering of these examples.

## 6.2 Evaluating on Mislabeled Data

In this subsection, we examine the impact of mislabeled data in evaluation sets and its potential to distort results. Labeling errors can mislead the evaluation process, resulting in inaccurate performance metrics and, in some cases, flawed model comparisons that lead to incorrect conclusions.

**Experimental Setup.** To test this assumption, we evaluate the performance of nine models, mostly state-of-the-art LLMs, on the test datasets. We compare their performance between the *Original* labels, and the *Gold* labels. For LLMs, we used zero-shot prediction as described in subsection 2.2, and averaged over prompts. For DeBERTa-based models we used the fine-tuned models from subsection 6.1, and averaged over seeds.

Table 4: Comparison of Model Performance on Original and Gold Labels. Ranking is defined over ROC AUC.

| Model | Rank | | ROC AUC | | F1 Score | | Accuracy | |
|---|---|---|---|---|---|---|---|---|
| | Original | Gold | Original | Gold | Original | Gold | Original | Gold |
| GPT-4 | 3 | 1 (+2) | 0.81 | 0.93 (+15%) | 0.73 | 0.83 (+14%) | 0.73 | 0.83 (+14%) |
| NLI model | 1 | 2 (–1) | 0.93 | 0.91 (-2%) | 0.87 | 0.87 (–) | 0.87 | 0.87 (–) |
| PaLM2 | 6 | 3 (+3) | 0.81 | 0.91 (+12%) | 0.71 | 0.81 (+14%) | 0.71 | 0.81 (+14%) |
| GPT-4o | 4 | 4 (–) | 0.81 | 0.91 (+12%) | 0.74 | 0.83 (+12%) | 0.74 | 0.83 (+12%) |
| GPT-4-mini | 5 | 5 (–) | 0.81 | 0.91 (+12%) | 0.71 | 0.79 (+11%) | 0.70 | 0.79 (+13%) |
| Llama3 | 7 | 6 (+1) | 0.75 | 0.86 (+15%) | 0.47 | 0.50 (+6%) | 0.52 | 0.55 (+6%) |
| Mistral-v0.3 | 8 | 7 (+1) | 0.75 | 0.85 (+13%) | 0.61 | 0.68 (+11%) | 0.62 | 0.68 (+10%) |
| DeBERTa-v3 | 2 | 8 (–6) | 0.84 | 0.80 (-5%) | 0.76 | 0.73 (-4%) | 0.76 | 0.73 (-4%) |
| Mistral-v0.2 | 9 | 9 (–) | 0.73 | 0.82 (+12%) | 0.66 | 0.72 (+9%) | 0.66 | 0.72 (+9%) |

**Results.** Prior to this work, an evaluation of these models would induce the values and ranking as in Table 4 under the *Original* sub-columns. However, as shown before, these datasets include labeling errors, and therefore do not support fair evaluation. Considering the new gold labels, based on expert

---

[3] `https://huggingface.co/microsoft/deberta-v3-base`
[4] `https://huggingface.co/MoritzLaurer/DeBERTa-v3-base-mnli-fever-anli`

intervention (as described in subsection 3.2), we obtain different results, shown in the *Gold* sub-columns. The first observed discrepancy is the ranking of models. For example, DeBERTa-v3 has shifted from being the second-best to the second-worst. Beyond the change in ranking, all metrics' (i.e., ROC AUC, F1-score and accuracy) range has shifted upward, indicating that LLMs perform better on this task than what was previously thought, likely due to label errors. If this phenomenon extends to other tasks and datasets beyond those examined in this study, it could suggest that LLMs are better than currently perceived.

## 7 RELATED WORK

### 7.1 LLMS IN THE ANNOTATION LOOP

LLMs have been increasingly utilized as annotators in various NLP tasks, offering potential benefits in efficiency and scalability. Several studies have demonstrated that LLMs can effectively generate annotations from scratch, sometimes outperforming human annotators or crowd workers (He et al., 2023; Gilardi et al., 2023; Törnberg, 2023). However, LLMs are not flawless and cannot be considered gold-standard annotators when used alone. They may produce incorrect annotations, especially in complex (Chen et al., 2024), social (Felkner et al., 2024), or low-resource (Bhat & Varma, 2023) contexts. These studies showed that LLMs can exhibit poor performance and biases, highlighting the necessity of human oversight to ensure quality or fairness. To address this issue, several approaches for collaborative (Kim et al., 2024; Li et al., 2023) or active learning (Zhang et al., 2023; Kholodna et al., 2024) were suggested, where LLMs and humans are both part of the annotation procedure.

### 7.2 HANDLING LABEL ERRORS

Label errors (also referred to as label noise) in training and evaluation datasets can significantly impair NLP model performance and reliability (Frénay & Verleysen, 2014). Fine-tuned models have been employed to detect mislabeled data by identifying examples with high loss or low confidence (Chong et al., 2022a; Hao et al., 2020; Pleiss et al., 2020; Northcutt et al., 2019). For example, Chong et al. (2022a) demonstrated that fine-tuned pre-trained language models can effectively detect label errors by ranking data points based on the training loss. Once these high-loss or low-confidence examples are flagged, they are typically filtered out (Nguyen et al., 2019; Northcutt et al., 2019), corrected automatically (Pleiss et al., 2020; Hao et al., 2020), or re-labeled by human annotators (Northcutt et al., 2021) to verify and improve dataset quality.

### 7.3 FACTUAL CONSISTENCY EVALUATION

Factual consistency evaluation (FCE) refers to the task of verifying that generated text remains true to the facts in the source content, addressing factual inaccuracies in models' outputs. It has been applied to tasks like summarization (Kryscinski et al., 2019; Xie et al., 2021; Gekhman et al., 2023) and dialogue (Honovich et al., 2021; Xue et al., 2023), which are prone to suffer from hallucinated outputs. Benchmarks like the TRUE Honovich et al. (2022) standardize evaluation across datasets. Common methods include entailment-based models (Laban et al., 2022) and QA-based approaches such as $Q^2$ (Honovich et al., 2021) and QAFactEval (Fabbri et al., 2021), with recent advancements like WeCheck (Wu et al., 2023) improving efficiency through weakly supervised learning.

## 8 DISCUSSION

Labeling errors are a persistent issue in NLP datasets, negatively affecting model fine-tuning and evaluation. Our findings demonstrate that LLMs, particularly when highly confident, can effectively detect these errors, outperforming crowd workers in accuracy, consistency, and cost-efficiency. As LLM capabilities advance, their role in refining data quality will become central to improving NLP benchmarks. Future work could explore applying LLM-based error detection to a broader range of datasets and tasks, as well as refining methods for optimizing label correction strategies. We encourage researchers to adopt our methods and critically evaluate existing datasets to drive more robust, reliable results in the field.

## ETHICS STATEMENT

We address several ethical considerations related to human annotators and the research community.

First, we recognize the significant human effort and cost involved in creating the datasets used in this study. While we question certain labels in these datasets, this should not be seen as undermining their value or the hard work behind them. These datasets have been highly beneficial to the research community, and our aim is to help improve labeling quality, especially as powerful tools like LLMs become more capable in various tasks. Our goal is to highlight areas where improvements can be made, contributing to further advancements in the field.

Additionally, we used crowd-sourced human annotators for text labeling. All participants were paid fairly, in line with platform regulations and our institution's policies. We ensured transparency in the process, treated participants with respect, and provided appropriate compensation for their efforts.

Lastly, we acknowledge the potential impact of LLMs on crowd-sourced workers who depend on these platforms for income. While we explore the use of LLMs to enhance or potentially replace certain aspects of annotation, we do not intend for this to harm human workers. Instead, we hope that crowd-sourced workers will adopt these tools, allowing them to become more efficient and skilled, which will improve both the scalability and quality of future datasets while maintaining a role for human oversight.

## REPRODUCIBILITY STATEMENT

In the study, we performed experiments involving human annotators, including both experts and crowd-workers. While the exact results of these experiments cannot be fully reproduced due to the inherent variability of human participants, the process can be replicated. To facilitate this, we provide detailed guidelines, platform setup, and technical specifications related to the annotation procedure of crowd-source (subsection 3.2, subsection B.1) and experts (subsection 3.2, subsection B.2).

For the experiments involving LLMs, we have included detailed procedural steps (subsection 2.2; subsection 3.2; and Appendix B.3), the prompts used (Figure 8), model versions (subsection 3.2; subsection 6.1; and Appendix D.2), hardware specifications and hyperparameter configurations (Appendix D.1), statistical measures (Appendix E), and evaluation metrics. These materials ensure that the LLM experiments are reproducible.

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

APPENDIX

# A THE POWER OF ENSEMBLE

As mentioned in subsection 3.2, we treat the LLM annotations as an ensemble of 2 models combined with 4 different prompts, in order to ensure greater stability in the results. Where one LLM may succeed, the other may fail, and averaging all their probabilities enables us to have more confidence in the final answer. In this subsection, we further analyzed the performance of LLMs by varying the size of the LLM ensemble, examining how this affects the model performance. We evaluate two aspects of model performance. First, we assess how closely the ensemble's annotations match the gold labels — essentially, how much we can trust the LLM annotations. We measure this aspect of label quality using the ROC AUC compared to the gold labels. The second aspect is the ensemble's ability to detect label errors. For this, we compute the F1-score by averaging the recall of errors and the precision of correctly identifying a candidate as a true error.

Results are shown in Figure 5. For both aspects, we see a clear trend. As we increase the number of models in the ensemble, the performance increases. In terms of ROC AUC w.r.t the gold labels (left plot), this suggests better annotation quality, while the right plot, a higher F1 score indicates a stronger error detector, either by recalling more errors or improving precision, or through a balance of both. Additionally, for both measures the variance decreases as the ensemble size grows, which indicates more stable and consistent annotations and error detections. Although not yet discussed in the context of error detection with LLMs, these results align with previous work showing the power of ensemble (Dietterich, 2007). These observations justify our choice to use an ensemble of models rather than a single one.

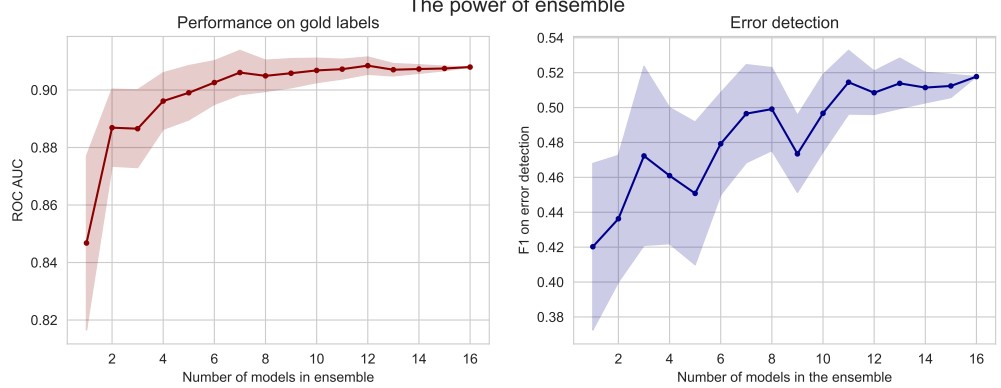

Figure 5: LLM Ensemble of different sizes (random subsets). **(Left)** presents the performance of the ensemble in terms of ROC AUC compared to the gold labels. **(Right)** presents the increasing ability to detect label errors. F1 is computed over *Error / Not Error* predictions.

# B ANNOTATION APPROACHES

## B.1 CROWD-SOURCE

### B.1.1 ANNOTATION PROCEDURE.

Each example was annotated by three annotators, that in addition to the binary label were requested to provide their confidence in their answer, and also write a short explanation for why they chose this label. Pre-qualifications included 50+ approved HITs, 97%+ approval rate, and locations from [USA, UK, Australia], which are all English-speaker countries. We disabled the possibility of right-click and `Ctrl+c` in the platform (as suggested by (Veselovsky et al., 2023a)), to prevent (as much as possible) the case where generative-AI (e.g., ChatGPT) will be applied to solve the task instead of humans solving it themselves (as shown by (Veselovsky et al., 2023b)). Maximum time allowed per HIT was 6 minutes, while the actual average execution time was 2:20 minutes for all assignments,

**Factual Consistency Evaluation - Instructions**                              ×

Thank you for participating in our research on factual consistency in texts.

Each example consists of two texts:

1. **Grounding** - A factual text.
2. **Statement** - A text to be evaluated.

**Task:**

Your task is to determine if the Statement is factually consistent with the Grounding.

**Definition of Factual Consistency:**

- **Factual Consistency:** The Statement accurately reflects and aligns with all the facts presented in the Grounding. The Statement does not introduce any errors, new entities, or unsupported information and is in full agreement with the Grounding.
- **Factual Inconsistency:** The Statement contains any inaccuracies, contradictions, or information that cannot be supported by the Grounding or derived from it.

**Answer Format:**

Your answer should be binary: either **Factually Consistent** or **Factually Inconsistent** (choose the appropriate answer in the "Your Answer" section).
Additional Information Required:

- Confidence Level: Indicate your confidence in your answer on a scale of 1 to 5 ("Your Confidence").
- Explanation: Provide a brief explanation for your answer ("Short Explanation" text box).

We appreciate your attention to detail and accuracy in this evaluation process. Thank you for your valuable contribution.

**Grounding:**

At the same time , Pope Francis Tong asked Bishop of Hong Kong to stay for three years .

**Statement:**

At the same time , Pope Francis asked Tong to remain Bishop of Hong Kong for three more years .

Your task is to determine if the Statement is factually consistent with the Grounding.

**Your Answer:**

○ Factually Inconsistent
○ Factually Consistent

**Your Confidence:**

Indicate your confidence in your answer on a scale of 1 to 5.
(Note: 0 is not part of the scale)

0

**Short Explanation:**

Provide a brief but meaningful explanation (at least one sentence) for why you classified the statement as factually consistent or inconsistent

Submit

Figure 6: Platform for crowd-sourcing annotation in Amazon Mechanical Turk (MTurk). **(Top)** Guidelines for the task and definitions. **(Bottom)** Annotation layout for a single instance.

and 3 minutes for approved assignments. The guidelines provided to annotators and the annotation platform layout are presented in Figure 6.

Each annotation was manually reviewed and was rejected if the answers were not in line with the instructions, or if it was obvious that the task wasn't done honestly. Overall, this task suffered from a high rejection rate of 49.2% (1163 rejected, 1200 approved). The main rejection reasons were: lack of meaningful explanation, obvious copy-paste annotations across different examples, explanation contradict with the label annotation, and cases where the explanation was a copy-paste of either the grounding or the statement.

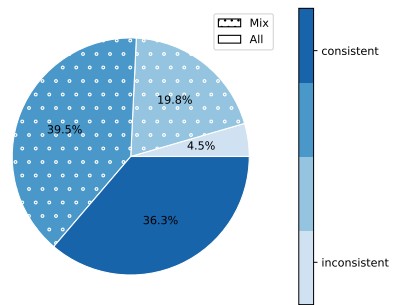

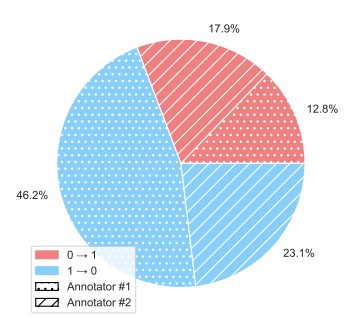

Table 5: Distribution of crowd-source annotators. Each example was annotated by 3 workers. Plain segments are unanimous annotation, while dotted segments indicate examples where some annotators labeled as *inconsistent*, and other as *consistent*. For example, 19.8% of the examples had two *inconsistent* annotation, and one *consistent* annotation.

Table 6: How experts annotations have changed after the reconciliation phase. Most changes occur from 1 (*consistent*) to 0 (*inconsistent*).

### B.1.2 CONSISTENCY

Crowd workers showed near-random agreement, indicating relatively poor-quality annotations. Table 5 describes the distribution of annotations by MTurk workers. Only 40.8% from the examples were labeled unanimously, where the rest included annotations from both classes. In addition, if aggregating as majority vote, we get that 75.8% of the examples are labeled as *consistent*, which is far from the original distribution of classes. As mentioned before, even experts may miss a small inconsistency nuance, and finding it requires attention. Even from the subset of examples unanimously labeled as *consistent*, 37.9% have a label of *inconsistent* in both original and gold labels, which points at a lack of attention and thoroughness.

### B.2 EXPERTS

Experts annotation was using the platform of Label Studio.[5] Layout design is presented in Figure 7. Examples were presented in random order, and neither the LLM prediction nor the original label were presented during the annotation. At the first stage, each example was annotated independently both experts. Afterwards, the human experts began in a second phase of a reconciliation, where a discussion was made over examples they disagree over. This reconciliation phase ended up with a much higher agreement, and higher-quality labels.

In the reconciliation phase, we observed that most changes (69.3%) were from label 1 to label 0, indicating that contradictions might be hard to find, and not all annotators catch them at first. For the full distribution of label change in the reconciliation phase, see Table 6.

### B.3 LLMs

We used four LLMs for the task of annotating a total of $160 \times 4 = 640$ examples from four different datasets. Each model was run with four different prompts (see full prompts in Figure 8). We used a variety of terminology, as this task appear with different framing in different studies. For example, the premise-hypothesis terminology from classic NLI (MacCartney & Manning, 2009), or document-statement used in (Tam et al., 2023).

For API models (GPT-4, PaLM2), we set `temperature=0.0` and extracted the logit of the generated token (functionality provided by both APIs), if the generated token was either `'0'` or `'1'` as expected. This logit was then transformed into a probability $p_t = P(y = t|x)$ via exponent corresponding the generated token $t$, and $1 - p_t$ for the other label. To address the case where the first generated token was an unrelated token such as `' '`, `'\n'`, we set `max_tokens=2` and took the first appearance of either `'0'` or `'1'`. For all models, prompts and examples, `'0'` or `'1'` were in the first two generated tokens. Rest of parameters were set according to their default values.

---

[5]https://labelstud.io/

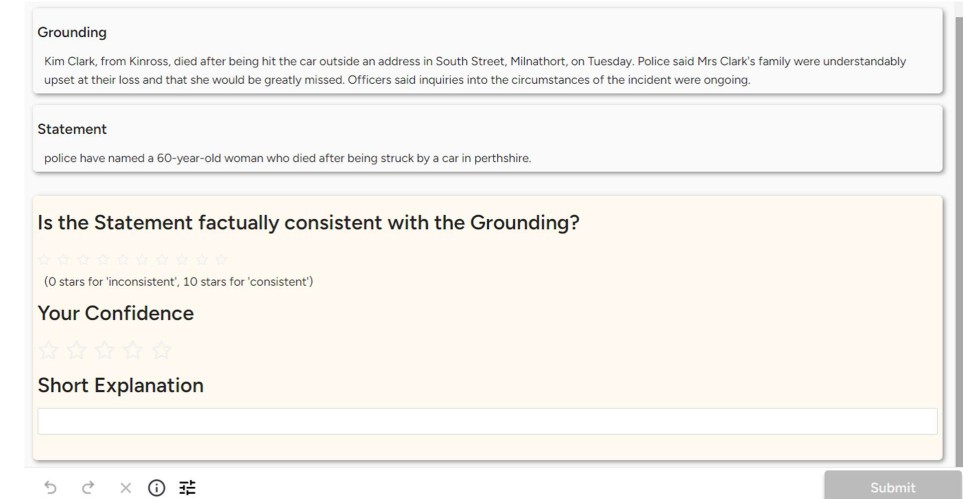

Figure 7: Annotation platform on Label-Studio for experts

For models available through the HuggingFace API (e.g., Mistral, Llama 3), we can load the model parameters and make inference locally. In that case, we get access to logits for all tokens, instead of just for the generated ones. Therefore, we applied a similar procedure, where we seek for the first appearance of either '0' or '1' to be the most probable token to be generated, and then directly extracted the logits of the '0' and '1' tokens. These logits were transformed into probabilities $(P(y=0|x), P(y=1|x))$ via a softmax function.

## C    COST AND SCALABILITY

In section 5 we compare the different annotation approaches on label quality, consistency and cost-effectiveness. Here are the full details regarding run-time and costs.

In MTurk platform, a total of $400 \times 3 = 1200$ annotations cost $572\$$, including 2 small pilot experiments. All annotations were prepared within a few hours. However, it demanded an additional and significant time for review, after which rejected examples returned to the pool. This annotation-review cycle was conducted for $\sim 5$ iterations. Inference via OpenAI's API on GPT-4 cost $\sim 4.5\$$ per prompt. Inference via VertexAI's API on PaLM2 cost $\sim 0.15\$$ per prompt. Both took $\sim 8$ minutes per prompt. Inference on Mistral and Llama3 was via the HuggingFace API, and its cost is estimated by the cost of using a suitable Virtual Machine (VM) on Google Cloud Platform (GCP) for the time of inference (1 minute per model)- $\sim 0.1\$$ per prompt. In total, annotating with an LLM is $\sim 10^2$ to $10^3$ times cheaper than a human annotator in a crowd-sourcing platform, and of course than human experts. It is also faster if considering reviewing procedure (excluding code writing or layout designing).

## D    MISLABELED DATA IMPLICATIONS

### D.1    FINE-TUNING

**Hardware.**  For the finetuning of DeBERTa models, both the base pre-trained model, and the NLI model which is in the same size, in subsection 6.1, we used 2 Quadro RTX6000 (24GB) GPUs.

**Implementation.**  We used HuggingFace trainer with early stopping of 4 epochs. The finetuning procedure includes splitting the training set into train and validation sets (where validation size is 25% and train 75%), fine-tuning on the train set, and choosing the best checkpoint based on the validation ROC AUC. We ran all experiments on five different seeds, affecting also the train-validation split and the random set chosen for ablation. We fine-tuned all variants with the same hyperparameters, determined by the best performing on the no-manipulation baseline. This includes

**prompt1**

Here are two texts:

TEXT 1. <..PREMISE..>.

TEXT 2. <..HYPOTHESIS..>.

Is TEXT 2 contradictory or is it factually inconsistent with TEXT 1? If yes answer 0.

Is TEXT 2 entailed or is it factually consistent with TEXT 1? If yes answer 1.

Refer only to the two texts above, and not any other external knowledge or context.

Answer only 0 or 1

Answer only with one token: 0 or 1

Answer:

---

**prompt2**

DOCUMENT: <..PREMISE..>.

QUESTION: Is the following STATEMENT factually consistent with the above document?

STATEMENT: <..HYPOTHESIS..>.

ANSWER FORMAT: 0 for No, 1 for Yes

Answer only with one token: 0 or 1

Answer:

---

**prompt3**

You are given the two following texts:

TEXT 1. <..PREMISE..>.

TEXT 2. <..HYPOTHESIS..>.

TEXT 1 is a fact. TEXT 2 is a statement. Is TEXT 2 factually consistent with TEXT 1?

Answer 0 for No, 1 for Yes.

Answer only with one token: 0 or 1

Answer:

---

**prompt4**

Given the following texts:

<PREMISE> : <..PREMISE..>.

<HYPOTHESIS> : <..HYPOTHESIS..>.

Please assess the factual consistency of <HYPOTHESIS> with respect to <PREMISE>.

If the content of <HYPOTHESIS> aligns with the information provided in <PREMISE>, assign a label of 1.

If there are factual inconsistencies between <HYPOTHESIS> and <PREMISE>, assign a label of 0.

Target Format: either 0 (for Factual Inconsistency) or 1 (for Factual Consistency).

Answer only with one token: 0 or 1

Answer:

Figure 8: Four different prompt input templates to LLMs for obtaining binary labels

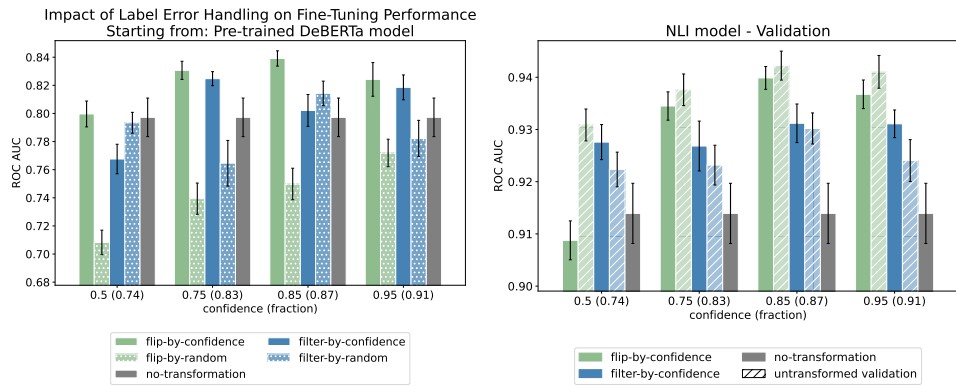

Figure 9: Similar experiments to the one in Figure 4, with small alterations. **(Left)** Starting from a different base model - pre-trained `DeBERTa-v3-base`. **(Right)** Dashed columns present results for when flipping or filtering methods were applied only on the training set, but not the validation.

30 epochs at most, batch size of 16, learning rate of `5e-5` and weight-decay of 0.03. The rest were set as the trainer and model default.

**Additional Experiments.** The left plot in Figure 9 presents the same experiment discussed in subsection 6.1, but starting from the pre-trained `DeBERTa-v3-base`. Same trends applies here, where our LLM-confidence-based manipulations of either flipping or filtering flagged examples outperforms the baselines.

The right plot in Figure 9 compares the performance of these methods (starting from the NLI model) when applied to both the training and validation sets (solid bars) or only the training set (dashed bars). The results are consistent, with no statistically significant differences between the two settings. Importantly, all variations outperform the baseline, underscoring the critical role of a well-curated training set in enhancing the model's ability to generalize effectively.

### D.2    MODEL EVALUATION

In subsection 6.2 we evaluated the following models: GPT-4, PaLM2 (`text-bison@002`), Mistral-v0.2 (7B), and Llama3 (8B), which are covered in subsection 3.2; DeBERTa-v3 and NLI-model, which is a fine-tuned version of it on NLI datasets, as discussed in subsection 6.1; and GPT-4o, GPT-4o-mini, Mistral-v0.3,[6] which share the same implementation as GPT-4 or Mistral-v0.2.

## E    STATISTICAL ANALYSIS

### E.1    CLOPPER-PEARSON

As mentioned in subsection 4.1, we we employed the Clopper-Pearson exact method (Clopper & Pearson, 1934) to construct a 95% confidence interval for the binomial proportion, adjusted by a finite population correction (FPC). As we only have a subset of examples we re-annotated by LLMs or experts, we can not precisely determine what is the error rate in the full dataset, but only construct a confidence interval based on the re-annotated subset. The Clopper-Pearson method provides an exact confidence interval for a binomial proportion, which means it gives a reliable estimate even with small sample sizes. By applying the finite population correction (FPC), we adjust the interval because our sample is drawn from a limited population. This adjustment helps refine the estimate by taking into account the size of the overall dataset compared to the sample.

For your reference, the sizes of the complete datasets are provided in Table 7.

---

[6]`https://huggingface.co/mistralai/Mistral-7B-Instruct-v0.3`

| Dataset | Task | % pos | Subset Size | Full Dataset Size |
|---------|------|-------|-------------|-------------------|
| MNBM | Summarization | 10.6 | 640 | 2500 |
| BEGIN | Dialogue | 38.7 | 640 | 836 |
| VitaminC | Fact Verification | 52.5 | 640 | 63504 |
| PAWS | Paraphrasing | 44.3 | 640 | 8000 |

Table 7: General datasets information.

### E.2 BOOTSTRAP SAMPLING

In subsection 4.1, we use bootstrap sampling to provide confidence intervals for each bin. Unlike the method in subsection E.1, we do not make claims about the entire dataset, but rather focus on the re-annotated subset we possess. To achieve this, we perform 100 bootstrap samples from the empirical distribution of each bin, sampling with replacement. We then measure the agreement between the experts' resolutions and the LLM annotations, compared to its agreement with the original label.

## F LABEL ERRORS

Table 8 demonstrates one example per dataset, in which the original label is in fact an error, the LLM prediction marked it as a candidate, and the expert annotators determined the correct gold label.

Table 8: Annotation errors in the original datasets, discovered by LLMs and corrected by experts.

---

**Dataset:** VITC

**Grounding:** The British Government and NHS have set up a Coronavirus isolation facility at Arrowe Park Hospital in The Wirral for British People coming back on a special flight from Wuhan. Evacuation of foreign diplomats and citizens from Wuhan. Due to the effective lockdown of public transport in Wuhan and Hubei province , several countries have started to evacuate their citizens and/or diplomatic staff from the area , primarily through chartered flights of the home nation that have been provided clearance by Chinese authorities.

**Generated Text:** There is a Coronavirus isolation facility at Arrowe Park Hospital that was set up by the NHS and the British Government

**Original Label:** 0      **LLM $p$:** 0.99      **Gold Label:** 1

Explanation: Rephrasing of the first sentence, without any contradiction.

---

**Dataset:** BEGIN

**Grounding:** Hillary Clinton, the nominee of the Democratic Party for president of the United States in 2016, has taken positions on political issues while serving as First Lady of Arkansas (1979–81; 1983–92), First Lady of the United States (1993–2001);

**Generated Text:** She is the nominee in 2016.

**Original Label:** 0      **LLM $p$:** 0.98      **Gold Label:** 1

Explanation: She (Hillary Clinton) is indeed the nominee in 2016 as specifically stated in the grounding.

---

**Dataset:** PAWS

**Grounding:** David was born in Coventry on 21 September 1933 , with his twin Charles and Jessamine Robbins , the eighth and ninth children of twelve by Robbins.

**Generated Text:** David was born on September 21 , 1933 in Coventry with his twin father Charles and Jessamine Robbins , the eighth and ninth child of twelve of Robbins

**Original Label:** 1      **LLM $p$:** 0.04      **Gold Label:** 0

Explanation: The generated text incorrectly states "twin father" instead of "twin" which is not the same, and does not even make much sense in English.

---

**Dataset:** MNBM

**Grounding:** The John Deere tractor was pulled over by officers in the village of Ripley and had two other males on board. The vehicle had been seen in nearby Harrogate at about 05:00 GMT with no headlights on. Police said the driver had no licence, was not insured and did not have permission from the tractor's owner. The vehicle was seized, with the three due to be interviewed by officers. Posting on Twitter, Insp Chris Galley said: "A strange end to a night shift. 15-year-old lad driving a tractor as a taxi for his drunk mates."

**Generated Text:** a 15-year-old boy has been stopped by police after being seen driving a taxi on a night taxi.

**Original Label:** 1      **LLM $p$:** 0.19      **Gold Label:** 0

Explanation: The generated text claims that the 15-year-old boy was "driving a taxi on a night taxi", contradicting the grounding in which it was claimed that the boy was driving a tractor as a taxi

