# OpenReview forum: "Are LLMs Better than Reported? Detecting Label Errors and Mitigating Their Effect on Model Performance"
_ICLR.cc/2025/Conference — Submitted to ICLR 2025_

### Official Review · Reviewer_AKsY · 2024-10-27

**Soundness:** 4
**Presentation:** 4
**Contribution:** 3
**Rating:** 8
**Confidence:** 4

**Summary:**

This paper evaluate the label errors in existing datasets and examine the reliability of different annotating approaches: expert, crowd-sourcing, and LLM-as-a-judge. The authors prompt LLM to predict the label of four datasets and asked experts to verify the prediction if it disagrees with the original label. The result shows a 10~40% of label errors in four selected datasets. In addition, LLMs' prediction highly align to experts' label when the confidence score of LLMs is high. The authors also hire crowd-workers to annotating a subset of samples, and shows that the annotations of crowd-workers have a lower inter-annotation agreement and a higher disagreement with labels annotated by LLMs and experts, suggesting that LLM-as-a-judge is a reliable and efficient alternative for large-scale annotation tasks. In the end, the authors introduced two approaches to mitigate the label errors in a dataset: by filtering those data points and by flipping the wrong labels. Experimental result shows that models' fine-tuned on datasets cleaned with both approaches achieve higher performance.

**Strengths:**

1. The paper is well written and easy to follow.
2. This paper shows that LLMs can be used to annotate data or identify label errors with a high agreement with experts, which can be a good way to mitigate the scalability issue of dataset curation.
3. The authors conducted extensive experiments, making their arguments convincing.

**Weaknesses:**

1. The approach and experiment are limited in binary classification tasks, specifically in factual consistency. The argument could be stronger if the authors conduct additional experiments in other types of task.
2. The experimental result of the crowd-sourcing part may be questionable. In Appendix B.1, the authors mentioned that the pre-qualifications of the MTurk task included 50+ approved HITs (line 1075). However, to get a higher quality result, people usually set the minimum number of approved HITs at 3000. This may explain why the authors got a high rejection rate of 49.2% and a very low Fleiss’s kappa value of the workers. As a result, the argument in Sec. 5 that the quality of crowd-sourcing annotation is insufficient may be too strong.

**Questions:**

1. Did you look into why the Fleiss’s kappa of crowd-sourcing is that low? To my understanding, Fleiss’s kappa is usually higher than 0.2, even for some subjective tasks (e.g. preference of responses). It is weird to me that the kappa value is only 0.074.

Typo:
line 422: alnoe -> alone
line 427: TThe -> The

---

> ### Author Response · Authors · 2024-11-19
> **Response (1) - reviewer AKsY**
>
> Thank you for your positive and thorough review and enlightening comments. We are happy to learn that you found our paper thorough and convincing. We hope you find answers to your questions and concerns below:
>
> **Binary labels:** We acknowledge that our experiments include only binary labeling tasks. Although the labeling scheme is designed for factual consistency, the underlying tasks are diverse, including summarization, dialogue, fact verification, and paraphrasing. Please note that our approach can be extended to multi-class datasets, as generally described in Subsection 2.2.
>
> **Approved HITs:** We agree that tightening the qualifications could have resulted in better annotators and, consequently, more reliable results. However, the scale of ~100+ approved HITs, combined with a high ~95%+ approval rate, is quite standard [1, 2, 3]. We will add these citations to the paper to support our choice.
>
> **Fleisses Kappa:** First, we double-checked the calculation and confirmed its validity. As to why it has a low score, additional evidence for low agreement between annotators is shown in Figure 5, where there was little consensus on most examples. Furthermore, Figure 2 shows a low F1 score compared to other annotation methods, indicating the poor quality of the annotations. This low quality is reflected in both the accuracy of the annotations and their consistency (IAA). We believe other factors such as task difficulty or annotator fatigue may also contribute to these results.
>
> Thank you for your meticulous reading, which uncovered these typos. We will, of course, fix them.
>
> **References:**
>
> [1] Kazai, Gabriella, J. Kamps and Natasa Milic-Frayling. “An analysis of human factors and label accuracy in crowdsourcing relevance judgments.” *Information Retrieval* 16 (2012): 138 - 178.
>
> [2] Hauser, David N., Aaron J. Moss, Cheskie Rosenzweig, Shalom N. Jaffe, Jonathan Robinson and Leib Litman. “Evaluating CloudResearch’s Approved Group as a solution for problematic data quality on MTurk.” *Behavior Research Methods* 55 (2021): 3953 - 3964.
>
> [3] Chmielewski, Michael and Sarah C. Kucker. “An MTurk Crisis? Shifts in Data Quality and the Impact on Study Results.” *Social Psychological and Personality Science* 11 (2019): 464 - 473.

---

> > ### Comment · Reviewer_AKsY · 2024-11-27
> >
> > Thanks for the reply. You've answer all my questions.

---

### Official Review · Reviewer_F6Ph · 2024-11-03

**Soundness:** 3
**Presentation:** 3
**Contribution:** 2
**Rating:** 6
**Confidence:** 4

**Summary:**

This paper evaluates the potential of LLMs as tools to detect label errors in NLP datasets, comparing LLM-based annotations with crowd-sourced and expert annotations. Using four datasets from the TRUE benchmark, the authors aim to demonstrate that LLMs can effectively flag mislabeled instances and, by extension, improve the overall accuracy of model training and evaluation.

**Strengths:**

The paper provides a detailed empirical study that underscores the prevalence of labeling errors in commonly used NLP datasets. It is thorough in its methodological approach, with comprehensive experiments and multiple evaluation metrics. The use of LLM ensembles to assess annotation quality and error rates highlights the potential cost-effectiveness of this approach compared to traditional methods.

**Weaknesses:**

This paper lacks substantial innovation, as its primary contribution—using LLMs for label error detection—builds on existing methods without introducing a significant advancement. Similar approaches have already been explored in previous studies, and the work here does not establish a clear differentiation or improvement over these. Additionally, the paper could benefit from expanding its dataset scope and addressing challenges such as multi-class labels or ambiguous labeling scenarios. Without these considerations, the findings may not generalize well to other, more complex datasets.

**Questions:**

- Could the methodology be extended to detect errors in multi-class datasets or cases with label ambiguity (fuzzy label)?
- What plans do the authors have to validate this approach beyond the datasets examined?

---

> ### Author Response · Authors · 2024-11-19
> **Response (1) - reviewer F6Ph**
>
> Thank you for your detailed review. We hope you find answers to your questions and concerns below:
>
> **Similar approaches:**
> We respectfully disagree with the assertion that our work lacks substantial innovation. Our research offers a comprehensive and thorough investigation of label errors, addressing both detection and mitigation strategies using LLMs.
>
> We would like to highlight key distinctions between our method and similar approaches. While there is extensive literature on detecting label errors for fine-tuned models, most of these studies focus on detection within the training loop [1, 2, 3], which is not directly applicable to LLMs when fine-tuning is not feasible. Second, most existing work involving LLMs identifies problematic examples by analyzing model uncertainty, typically characterized by high entropy, low confidence in predictions, or external verifiers [4, 5, 6]. Our approach examines the confidence in the incorrect label, but only in cases where the original label disagrees with the prediction. Lastly, unlike prior work, our method leverages an ensemble of LLMs and prompt designs to enhance robustness [7, 8].
>
> Additionally, our contributions extend far beyond the methodological novelty (which constitutes only _a small part of our paper_) to include a comprehensive end-to-end analysis of detecting label errors, addressing and re-annotating them, and mitigating their impact on downstream tasks.
>
> First, we demonstrate that as LLM confidence increases, their precision in flagging mislabeled examples also rises. This confidence-based approach provides a more nuanced understanding of LLM reliability in error detection, which has not been explored in prior research.
>
> Furthermore, our extensive empirical investigation into the impact of label errors reveals the significant distortion these errors can cause in model performance. By correcting mislabeled data, we illustrate that performance can improve by up to 15%, and we show that many LLM-reported mistakes are, in fact, errors in the original labels. This finding challenges assumptions made in previous evaluations and emphasizes the need to reassess established benchmarks. We also address how to mitigate the negative effects of label errors during model fine-tuning. Our proposed methods, including filtering and label flipping based on LLM confidence, yield up to a 4% improvement in model performance. This practical approach highlights how our strategies can make models perform better and be more robust.
> Lastly, our work contributes significantly to the broader understanding of employing LLMs in annotation tasks. We comprehensively compare LLM-based error detection with traditional methods, revealing both strengths and limitations. By doing so, we provide a well-rounded analysis that informs future research and offers new ways to improve data quality.
>
> **Multi-class or ambiguous labeling scenarios:**
> Our general approach, as described in Subsection 2.2, is not limited to binary labeling and can be extended to multi-class datasets. We specifically chose datasets from the TRUE benchmark to explore various domains and tasks while benefiting from a unified labeling schema, which provided consistency and comparability across our experiments. While we did not explicitly address fuzzy labeling scenarios in this paper, we believe our approach could be adapted to handle such cases by defining terms like “disagree” or “strongly disagree” with more nuance (e.g., using KL-divergence between the label distributions to quantify the disagreement level).
>
> **Validation of our approach:** Our research includes extensive experiments that validate our method across multiple datasets, covering various tasks and domains. While we acknowledge that using more datasets would strengthen our findings, budget constraints played an important role, given the need to obtain annotations from multiple LLMs, crowd-workers, and experts. We hope that, once published, our study will inspire and motivate other researchers to apply and extend our approach in diverse scenarios (e.g., exploring other datasets, tasks, or, as you suggested, handling multi-class and ambiguous labeling cases), further validating and building upon our work.

---

> ### Author Response · Authors · 2024-11-19
> **Response (2) - reviewer F6Ph**
>
> **References:**
>
> [1] Chong, Derek, Jenny Hong, and Christopher D. Manning. "Detecting label errors by using pre-trained language models." arXiv preprint arXiv:2205.12702 (2022).
> ‏
>
> [2] Pleiss, Geoff, et al. "Identifying mislabeled data using the area under the margin ranking." Advances in Neural Information Processing Systems 33 (2020): 17044-17056.
> ‏
>
> [3] Swayamdipta, Swabha, et al. "Dataset cartography: Mapping and diagnosing datasets with training dynamics." arXiv preprint arXiv:2009.10795 (2020).‏
>
>
> [4] Huang, Hsiu-Yuan, et al. "A Survey of Uncertainty Estimation in LLMs: Theory Meets Practice." arXiv preprint arXiv:2410.15326 (2024).
> ‏
>
> [5] Klie, Jan-Christoph, Bonnie Webber, and Iryna Gurevych. "Annotation error detection: Analyzing the past and present for a more coherent future." Computational Linguistics 49.1 (2023): 157-198.
> ‏
>
> [6] Wang, Xinru, et al. "Human-LLM collaborative annotation through effective verification of LLM labels." Proceedings of the CHI Conference on Human Factors in Computing Systems. 2024.‏
>
>
> [7] Kholodna, Nataliia, et al. "Llms in the loop: Leveraging large language model annotations for active learning in low-resource languages." Joint European Conference on Machine Learning and Knowledge Discovery in Databases. Cham: Springer Nature Switzerland, 2024.
> ‏
>
> [8] Zhang, Ruoyu, et al. "Llmaaa: Making large language models as active annotators." arXiv preprint arXiv:2310.19596 (2023).‏

---

> > ### Comment · Reviewer_F6Ph · 2024-11-24
> >
> > Thanks for the response and clarification, I have improved my rating.

---

### Official Review · Reviewer_KjaH · 2024-11-03

**Soundness:** 1
**Presentation:** 4
**Contribution:** 2
**Rating:** 5
**Confidence:** 4

**Summary:**

The paper examines mislabeled instances in existing standard label-based datasets. The proposed method uses large language models (LLMs) to identify these instances, flagging them as potential errors when LLM predictions differ from the ground truth label. These flagged instances are then reviewed by expert human annotators for relabeling. Depending on the research question being answered, the paper compares LLM predictions and crowd-sourced annotations with expert annotations to identify mislabeled data and conduct further analysis. The findings reveal significant disagreement between expert and crowd-sourced annotations, resulting in a high rate of mislabeled data. Notably, LLM predictions align more closely with expert annotations.

**Strengths:**

1. The paper is well-written.
2. The topic is interesting, and the paper addresses important questions.
3. The approach used in the paper is clear and aligns well with the research questions mentioned.

**Weaknesses:**

1. The experiments were conducted with a small sample size and a small number of datasets. While it is understandable to control the budget when using LLMs and crowd-sourced annotators, I believe the sample size and the number of datasets in this study are too small to support the conclusions drawn. This is because a change in the label of even a few samples could lead to a significant shift in performance, and the central argument of this paper relies on this performance change.

2. In line with the previous comment, I am uncertain about the reliability of the results reported in Figure 2 and Table 3. If crowd-sourcing performs this poorly in matching expert annotations (gold labels), how did LLMs learn to predict labels that align more closely with expert annotations, given that they were trained on standard datasets labeled through crowd-sourcing?

3. According to the paper, the authors used their own guidelines to relabel dataset instances for both crowd-sourced and expert annotators. However, these guidelines differ from the original guidelines used to label the datasets. This creates a major discrepancy, as it raises the question of whether the newly assigned labels truly reflect the same conditions outlined in the original guidelines for determining ground truth labels.

4. While there is demographic information about the crowd-sourcing participants, such as their status as native English speakers, there is no such information available for expert annotators.

**Questions:**

**Question:**
1. The paper claims that LLMs show considerable agreement with expert annotators in a zero-shot setting for mislabeled samples. How do the authors justify this claim, given that it is widely known that performance improves in a few-shot setting? Specifically, if LLMs outperform crowd-sourced annotators in matching expert annotations, a decrease in performance should be expected. This is because LLMs, by learning from in-context examples, would align more closely with expert annotations, resulting in greater disagreement with crowd-sourced annotations.

**Suggestions:**
1. In line 79, it would be clearer to provide a numerical value instead of a verbal description.
2. The specific checkpoint used for GPT-4 is not mentioned in the paper.

---

> ### Author Response · Authors · 2024-11-19
> **Response (1) - reviewer KjaH**
>
> Thank you for your thorough response, we are happy you found our paper interesting and clear. We wish to address your concerns:
>
> **Sample size and the number of datasets:**
> We annotated examples by multiple LLMs, crowd-workers, and experts, and as you mentioned, budget control had to be considered. However, we disagree that the number of examples or the number of datasets is too small to make reliable conclusions. Please note that, when possible, for all analyses and experiments in the paper, we provide confidence intervals or confidence-based lower bound estimations that our conclusions take under consideration (e.g., Figure 1 and lines 275-278, Figure 4, Table 2). For example, in Figure 1, for the range [0.99,1.0] even the lower bound of the CI is above the 50% line, suggesting that for this bin of examples, we are certain that the LLM is correct more than the original label. In Figure 4 the whiskers show that the differences between the methods cannot be attributed to chance. In Table 1, we present a lower bound estimate of the error rate across the entire dataset (lines 247-253), and even at this conservative estimate, the error rate remains substantial.
> Although the sample size might be considered small, our use of robust statistical practices ensures the reliability and validity of the findings and conclusions.
> Additionally, our datasets span over domains and tasks, which supports the generalization of our findings across different contexts.
>
> **Crowd-source performance:** It is common for crowd-sourced annotators not to be employed in their basic form (i.e., untrained, common crowd-workers). Typically, these annotators undergo a selection process involving qualification tests and targeted training to improve their skills for the task and experience. Indeed, our findings, as shown in Figure 3, support that more experienced annotators provide better-quality labels.
> However, these trained workers fall on a spectrum somewhere between plain crowd-workers and expert annotators, rather than representing typical crowd-sourcing. Our paper focuses on the edges of that spectrum, using untrained, plain crowd-workers who do not receive task-specific training; nevertheless, note that they still meet basic pre-qualifications, such as having a history of approved HITs and a minimum approval rating. This likely explains the discrepancy noted. We stand behind our collection process (as fully detailed in subsection 3.2 and Appendix B.1), and therefore by the results it produces (e.g., in Figure 2 and Table 3), which show relatively poor performance from crowd-sourcing.
>
> On why LLMs align more closely with expert annotations despite training on crowd-labeled data– LLMs are trained on diverse datasets including expert-labeled data, data annotated by trained crowd-workers, and a substantial amount of unsupervised text. This broad exposure enables LLMs to generalize effectively and recognize subtle patterns, leading to a closer alignment with expert labels. In contrast to open-source models, which are trained primarily on publicly available datasets, closed-source models are more likely to optimize their training data through data cleaning and constructing custom, expert-annotated datasets.
> Our findings further illustrate this: experienced crowd-workers achieve an F1 score of approximately 0.73 (Figure 3), on par with open-source LLMs that reach around 0.7, while closed-source LLMs perform even better, obtaining an F1 score of ~0.8 (Table 4). This pattern supports your hypothesis: as LLMs rely more on crowd-sourced labels, their performance decreases, aligning less with expert annotations. This underscores our paper's emphasis on the crucial role of data quality.
>
> **Guidelines:**
> Upon manually examining mislabeled examples, we can confidently say that most label mismatches originate from errors, not guideline discrepancies. For example, see the annotations in Table 8. The label errors we identified are clear errors and do not stem from any potential misinterpretation of guidelines.
>
> Yet, we acknowledge that some minor discrepancies might exist between the original guidelines and the merged ones, and we will add this as a limitation in the discussion. However, our datasets are taken from a widely used benchmark for factual consistency - TRUE. As such, they all share a common meaning for labels (i.e., “consistency” should mean the same). When using the TRUE benchmark for evaluating models, the same prompt is used for every dataset, or the same fine-tuned model is evaluated on each dataset.
>
> Therefore, having unified guidelines, even if they were not explicitly established before, is reasonable, valid, and should not create major discrepancies from the original separate guidelines.
> We made efforts to stay true to the core characteristics and intentions of the original guidelines and took a strict approach, as described in TRUE. This is reflected in our careful use of terms like “all” or “any” in our instructions.

---

> ### Author Response · Authors · 2024-11-19
> **Response (2) - reviewer KjaH**
>
> **Demographic information:** We acknowledge the importance of such information and will gladly provide it in a camera-ready version if accepted. We ensure our expert annotators are experienced and knowledgeable in the field.
>
> **Few-shot:** Our claim that LLMs show considerable agreement with expert annotators in a zero-shot setting for mislabeled samples is supported by our findings, which demonstrate that zero-shot performance is already strong. We acknowledge that few-shot learning generally improves performance, and we believe this would also be true in our case, even if the few-shot exemplars contain some label errors. Several factors contribute to this improvement: few-shot exemplars are often carefully selected, which reduces the likelihood of errors; they are also primarily used to help the model adjust to the domain, context, and format of the task rather than to explicitly teach labeling rules [1, 2]. Moreover, even if the error rate is not zero, most data is correctly labeled, further aligning the model's predictions with expert annotations.
>
> Also, thank you for your suggestions, we will include them in our next revision.
>
> We hope you find answers to your concerns and questions in our response. Should any questions arise, we would be happy to respond.
>
> **References:**
>
> [1] Sewon Min, Xinxi Lyu, Ari Holtzman, Mikel Artetxe, Mike Lewis, Hannaneh Hajishirzi, and Luke Zettlemoyer. 2022. Rethinking the Role of Demonstrations: What Makes In-Context Learning Work?. In *Proceedings of the 2022 Conference on Empirical Methods in Natural Language Processing*, pages 11048–11064
>
> [2] Cheng, Chen, Xinzhi Yu, Haodong Wen, Jinsong Sun, Guanzhang Yue, Yihao Zhang and Zeming Wei. “Exploring the Robustness of In-Context Learning with Noisy Labels.” *ArXiv* abs/2404.18191 (2024): n. pag.

---

> ### Comment · Reviewer_KjaH · 2024-11-30
>
> I would like to thank the authors for their responses. However, my concerns remain unaddressed for the following reasons:
>
> >  Sample size and the number of datasets
>
> Regarding the robustness of the statistical tests you performed, since all such tests rely on estimations, the conclusions drawn heavily depend on the _sample size_ used. If the sample is too small, it does not yield meaningful conclusions, _regardless of the strength of statistical test employed._
>
> > Crowd-source performance
>
> With due respect, I find your argument unconvincing regarding the statement:
>
> _"LLMs are trained on diverse datasets, including expert-labeled data, data annotated by trained crowd-workers, and a substantial amount of unsupervised text. This broad exposure enables LLMs to generalize effectively and recognize subtle patterns, leading to a closer alignment with expert labels."_
>
> Your argument is not accurate for two reasons:
>
> 1. A significant proportion of the datasets are labeled via crowd-sourcing, which outweighs datasets labeled by experts. Therefore, it is more likely that models learn predominantly from crowd-sourced labeled data, which, according to your argument, are not of high quality and models cannot learn the mentioned nuances from these datasets.
>
> 2. The purpose of unsupervised learning is to capture general patterns, not task-specific patterns. If your claim were accurate, we would not need supervised fine-tuning or other alignment-focused training paradigms, as the model would already be capable of learning even subtle nuances through unsupervised learning alone. Clearly, this is not the case.
>
> > Guidelines
>
> While it is true that any dataset may contain mislabeled data, as you illustrated in Table 8, making broader claims requires strict adherence to consistent guidelines. Without such guidelines, a majority of mislabeled data could be misinterpreted as genuine errors, rather than being true mislabels.
>
> > Few-shot
>
> I did not mean using mislabeled demonstrations in in-context learning (ICL). My point was that, in few-shot ICL, where demonstrations are added randomly (considering the simplest case), performance improvements are commonly observed. __These improvements are measured based on labels provided by crowd-sourced annotators.__ __However, according to your results, as LLMs outperform crowd-sourced annotators in aligning with expert annotations, a decrease in performance should be expected due to the discrepancy you identified between crowd-sourced and expert annotators.__ However, this is not the case, as few-shot ICL typically enhances performance in most scenarios. How do you explain this apparent contradiction?

---

> > ### Author Response · Authors · 2024-11-30
> >
> > Thank you for your response. We wish to address your remaining concerns:
> >
> > **Sample size and the number of datasets:**
> >
> > We agree that statistical tests rely on estimations, as they are inherently probabilistic. However, our analyses employ a standard 95% confidence level, which is widely regarded as a robust criterion for making reliable conclusions. Importantly, these tests inherently account for sample size: while smaller sample sizes may yield broader confidence intervals, the conclusions drawn remain statistically valid within this framework. In our study, even with broader confidence intervals due to the sample size, the intervals consistently support our findings.
> >
> > While we acknowledge the sample size limitations, our use of datasets spanning multiple domains and tasks further mitigates concerns about generalizability. Together, the confidence intervals and the diversity of the datasets provide strong evidence for the reliability and applicability of our conclusions, even if a larger sample size could yield narrower intervals.
> >
> > We understand your concern and believe that *how many examples stand for a “big enough” sample size* may be a subjective matter. We respect that our perspectives may differ and are willing to agree to respectfully disagree. Nevertheless, this does not call into question the validity of the statistical conclusions.
> >
> >
> > **Crowd-source performance:**
> >
> >  We did not claim that expert-labeled datasets dominate LLM training data, nor that unsupervised learning alone is sufficient for achieving task-specific performance; yet, we can not ignore their contribution. Instead, our argument emphasizes the breadth and diversity of the training data, which include a mix of expert-labeled, crowd-sourced, and unsupervised datasets. While unsupervised learning primarily captures general patterns rather than task-specific nuances, these patterns provide a foundational understanding of language and context, enabling the model to generalize effectively and perform well in zero-shot or few-shot settings (e.g., [3, 4]). Supervised fine-tuning and alignment training then built upon this foundation, refining the model’s task-specific capabilities and enhancing its performance across a wide range of tasks.
> >
> > As noted in our previous response, crowd-sourced datasets span a spectrum—from untrained, common crowd-workers to carefully selected and trained annotators. Our paper focuses on the lower end of this spectrum, which explains the observed discrepancies. It is important to clarify that we do not claim all crowd-sourced datasets are of poor quality or suffer from high error rates. Rather, the quality depends significantly on the annotation process, and our findings reflect this variability.
> >
> > Finally, in our previous comment, we addressed your concern regarding training with crowd-sourced data by presenting results from our paper. Our findings indicate that experienced crowd workers perform on par with open-source LLMs, which are likely trained on a significant proportion of crowd-sourced data. However, they are outperformed by closed-source LLMs, built by companies like OpenAI and Google, that invest substantial resources to collect such high-quality training data. Therefore, we believe they benefit from training on higher proportions of domain-expert data or, at the very least, data from highly trained crowd workers who perform similarly to domain experts..
> >
> >
> > **Guidelines:**
> >
> > We would like to emphasize again that, following our manual examination, the vast majority of errors we identified (as shown in Table 8) are genuine mislabels rather than differences caused by guideline discrepancies.
> > Yet, we acknowledge that minor differences between the original and unified guidelines might exist, and thus, we will explicitly include this as a limitation in the paper.
> >
> > However, we stand by our claim that the unified guidelines are representative of the task and consistent across the datasets in the TRUE benchmark, which inherently assumes a shared definition of “consistency.” Our approach ensured coherence across datasets while remaining true to the benchmark’s core intentions. The careful construction of our guidelines, including precise terms like “all” or “any,” was designed to minimize subjectivity and maintain alignment with the original guidelines.

---

> > > ### Author Response · Authors · 2024-11-30
> > >
> > > **Few-shot:**
> > >
> > > If we understand correctly, you refer to the simplest case where few-shot exemplars are randomly chosen and are crowd-sourced annotated. In this case, two possible scenarios can occur: (1) all exemplars are correctly labeled, in which case performance improvement is expected as the model aligns more closely with the task; (2) some exemplars are mislabeled. Even in the latter scenario, as we noted in our initial response, the majority of exemplars are typically correct, which helps guide the model toward the appropriate relationship between examples and labels. Furthermore, mislabeled exemplars still provide useful context for understanding the domain and task format, rather than explicitly teaching the model incorrect labeling rules (e.g., see the papers cited in our previous comment).
> > >
> > > This aligns with the general observation that few-shot ICL improves performance, even when the exemplars are not perfect. The model’s ability to generalize and extract the relevant task context from the exemplars mitigates the potential impact of label noise, which explains why few-shot ICL can still enhance performance despite the discrepancy between crowd-sourced and expert annotations.
> > >
> > > \
> > > We hope our response answers your remaining concerns.
> > >
> > > **References:**
> > >
> > > [3] Takeshi Kojima, Shixiang Shane Gu, Machel Reid, Yutaka Matsuo, Yusuke Iwasawa:
> > > Large Language Models are Zero-Shot Reasoners. NeurIPS 2022
> > >
> > > [4] Tom B. Brown, Benjamin Mann, Nick Ryder, Melanie Subbiah, Jared Kaplan, et al.:
> > > Language Models are Few-Shot Learners. NeurIPS 2020

---

> > > > ### Comment · Reviewer_KjaH · 2024-12-03
> > > >
> > > > I would like to thank the authors for their responses. However, their replies largely repeat what was stated earlier. For instance, regarding the few-shot ICL, I clearly mentioned that I was not referring to including mislabeled demonstrations in the input prompt. My main point was focused on the improvement in performance. While I am inclined to lower my score, I have decided to keep it as it is.

---

### Author Response · Authors · 2024-11-24
**Dear reviewers - Please consider our responses**

Thank you all for your thoughtful and constructive reviews. We are pleased that Reviewers AKsY and KjaH found the paper well-written and clear. Additionally, Reviewers F6Ph and AKsY highlighted the detailed empirical study and extensive experiments, which convincingly underscore the importance of addressing labeling errors in NLP datasets. Both reviewers also emphasized the applicability of our proposed method, particularly its scalability and cost-effectiveness.

As the discussion period comes to an end, we kindly ask the reviewers to consider our responses to their reviews, which we hope have addressed any misunderstandings or concerns.

Thank you.

---

### Meta-Review · Area_Chair_iPVv · 2024-12-17

**Metareview:**

This paper investigates the use of LLMs to identify mislabeled instances in NLP datasets, comparing LLM predictions, expert annotations, and crowd-sourced labels. The study highlights the prevalence of labeling errors (10–40%) in four binary classification datasets and demonstrates that LLM predictions align more closely with expert annotations than crowd-sourced labels. While the findings are intriguing and the paper is well-written, several weaknesses limit its contribution. The study’s scope is narrow, focusing solely on binary classification and factual consistency tasks, which undermines the generalizability of the conclusions. Additionally, the small sample size and limited number of datasets reduce the robustness of the results, as even minor changes in annotations could significantly impact the findings. The experimental design also raises concerns, particularly regarding crowd-worker qualifications (low pre-qualification thresholds may have led to unreliable annotations) and the mismatch between relabeling guidelines and original dataset standards. Furthermore, the paper lacks innovation, as the approach of using LLMs for label error detection is incremental and builds on existing work without introducing significant methodological advancements. Given these limitations, the study falls short of providing sufficient novelty and generalizable insights to warrant acceptance.

**Additional Comments On Reviewer Discussion:**

The major discussion during the rebuttal phase is regarding the significance of the experimental results (e.g., mall sample size and limited number of datasets) and the validity of the annotation process, both were not well addressed during the rebuttal phase (quoting from Reviewer KjaH: "their replies largely repeat what was stated earlier"). Also, multiple reviewers pointed out that the setting only focuses on binary classification, which might not generalize to multi-class classification problems.

---

### Decision · Program_Chairs · 2025-01-22

Reject